# Designing Urban Green Infrastructures Using Open-Source Data—An Example in Çiğli, Izmir (Turkey)

**Stefano Salata** [1],[*] 🆔, **Bensu Erdoğan** [2] 🆔 and **Bersu Ayruş** [2] 🆔

[1] Izmir Institute of Technology, Laboratory of Ecosystem Planning and Circular Adaptation—Lab EPiCA, Department of City and Regional Planning, Gülbahçe Kampüsü Urla, Izmir 35430, Turkey

[2] Izmir Institute of Technology, Department of City and Regional Planning, Gülbahçe Kampüsü Urla, Izmir 35430, Turkey; bensuerdogan@std.iyte.edu.tr (B.E.); bersuayrus@std.iyte.edu.tr (B.A.)

[*] Correspondence: stefanosalata@iyte.edu.tr; Tel.: +90-0531-013-2269

**Abstract:** The city of Izmir (Turkey) has experienced one of the most rapid and fastest urbanization processes in the last thirty years; more than 33 thousand hectares of agricultural and seminatural land have been transformed into urban areas, leading to a drastic reduction of biodiversity and hard deployments of the ecosystem service supply. In this perspective, the potential definition of methodologies to design multifunctional green infrastructures is extremely important to challenge the effects of climate change. The aim of this study is to propose an easy and replicable methodology to design a Green Infrastructure at the neighbourhood level in one of the most important districts of Izmir: Çiğli. To this end, we combined historical land-use change analysis (based on Urban Atlas, Copernicus Land Monitoring Service) with environmental and ecosystem mapping in a Geographic Information System environment (ESRI ArcMap 10.8.1) while creating a composite layer based on unweighted overlays of Imperviousness, Tree Cover Density, and Habitat Quality. Results were used to design the Green Infrastructure of Çiğli and suggest context-based strategies for urban adaptation, including Nature-Based Solutions for core, edge, and urban links.

**Keywords:** green infrastructure; urban planning; GIS; ecosystem services; adaptation; environment

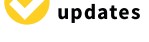



## 1. Introduction

Nowadays, scientists observe changes in the Earth's climate in almost every region of the World, and the entire system is undergoing a rapid transformation [1,2]. Climate change increases the possibility and the impacts of catastrophic environmental phenomena occurring at the ecological, societal, and economic level [3–5]. Most of the observed consequences are happening at an unprecedented rate [6], and perhaps these changes will be irreversible for hundreds of thousands of years [7,8]. Moreover, the increased frequency of hazards threatens the ecosystem's functioning, exacerbating the negative feedback on the microclimate and increasing the effects of extreme events [3,9]. Even the most banal problem of the ongoing sea-level rise or the loss of biodiversity will pose serious questions of adaptation, migration, and emargination of communities [10,11]. The one-meter prediction of rising sea levels with the ongoing emission trends will cause serious ecological, social, and economic damage to the coastal cities as well as many less densely inhabited coastal settlements [12,13].

The vulnerability to climate change varies according to urban spatial characteristics [14,15]. Many urban agglomerations are located by the sea or facing oceans (coastal areas). Coastal areas, for example, are overexposed to climate change: both sea levels rise and pluvial flooding, erosion, salinization, or storm inundation can impact their economic activities and trades; thus, the potential material, human, and moral losses that may be exposed can be really serious [12,16,17]. Unfortunately, these hazards often happen simultaneously or within a short time, thus calling for a real integrated and holistic approach to

their management [18–20]. Furthermore, coastal cities are growing quickly while exposing more of the population to threats and odds [21–23].

For the reasons mentioned above, the adaptation of these coastal systems should be be effective and should take action immediately since the process of transformation of densely inhabited urban systems typically needs a long time, and it should be based on a sound knowledge of the local environmental vulnerabilities [13,24]. İt has been demonstrated that more natural solutions can achieve the resilience of socio-ecological and technological systems to climate change [25]. However, the process of greening by urban forestry or de-sealing requires huge investments, coordination, and the application of adaptive planning principles on the urban agenda [26]. At the same time, adaptive planning can be achieved only by having a deep biophysical knowledge of coastal urban areas using the latest available technology while evaluating the current and, whereas possible, the future situation of the cities' ecosystems.

Green Infrastructure (GI), along with other environmental planning tools such as Urban Growth Boundaries, Net Environmental Benefit analysis, and costs for developing, represent an advanced approach to regulating urban expansion, limiting the land take, and increasing the resilience of urban areas [27,28]. The above-mentioned approaches support the application of different measures for re-development and become crucial in practical planning procedures [29].

In the European Commission context, GI implementation is considered a priority for urban adaptation. European institutions are fully committed to implementing GI as the primary element of their strategy to achieve a climate-free Europe and preserve natural habitats for the benefit of the ecosystem [30]. On 6 May 2013, the European Commission adopted an EU-wide strategy promoting investments in ecosystem services to prevent the increased risk of climate change and enhance Green Infrastructure (GI) in urban spatial planning. According to the EU, Ecosystem Services (ESs) are the free benefits that flow from nature functioning to people [31–33]. They can be categorized as provisioning (e.g., the supply of food, clean air and water, and materials), regulating (e.g., water and climate regulation, nutrient cycling, pollination, or the formation of fertile soils), or cultural (e.g., recreation opportunities, or the inspiration we draw from nature) [34,35]. On the other hand, GI is a strategically planned network of natural and semi-natural areas with other environmental features designed and managed to deliver a wide range of ESs, such as water purification, air quality, space for recreation, climate mitigation, and adaptation [36,37]. The idea behind the growing interest on these issues is that investment in GI is based on the logic that the costs of NBS will always be more profitable than replacing these environmental services with human technological solutions [38,39]. In addition, the avoided damage costs that GI can mitigate will greatly overcome the initial costs for their implementation [40,41]. Indeed, GI is nowadays considered the basic Supporting decision making system to reduce climate change vulnerabilities in different areas of the world [42,43].

GI and Nature-Based Solutions are considered a basic strategy to decrease the vulnerability of cities to extreme events and, at the same time, represent the basic conditions to achieve more resilient socio-ecological and technological systems [44–46]. Nature-based Solutions (NBS) is an umbrella concept covering a range of ecosystem-based approaches, including protection, sustainable management, restoration, and creation of natural or green infrastructure [24,47,48]. These approaches can be considered in a hierarchy prioritizing the preservation of existing ecosystems over enhanced management, rehabilitation and restoration, or the creation of new NBS [49]. Nevertheless, to be effective, NBS should be designed according to some performance-based principles that recognize the basic fundamental landscape ecology approaches that distinguish the core, edge, and connection of ecological areas in a system [50,51].

Basically, GI embraces actions of preventing climate change as it is designed and managed to improve the overall ecological quality of the broader urban system (e.g., reducing flood risk, reducing carbon emissions, reducing the urban heat island effect, disaster prevention), ensuring in the long-term the maintenance and increase in biodiversity

(e.g., enhanced habitats for wildlife, ecological corridors, landscape integrity, and ecosystem reconditioning) [52–54]. Still, their efficacy mostly relies on the biophysical knowledge of the environmental assessment of the region, city, or district where these infrastructures are designed. Recent advancements in GI design demonstrates how the evaluation of the multifunctional character of urban green areas should be based on spatial models to map the ecosystem delivering capacities of different lands parcels simultaneously [37].

Multifunctionality represents an advancement of the traditional landscape ecology approach which has been applied around the concept of ecological processes and ecological unbalances. Nowadays, the multifunctional ESs assessment is designed to emphasize the different benefits (combining supporting, regulation, provisioning, and cultural ecosystems) that Natural Capital can play in increasing the quality of life of citizens, including immaterial benefits derived by the aesthetic values of urban green [55,56]. ESs modelling is one of the most effective human technological solutions for environmental planning and offers possibilities for understanding the socio-ecological and technological systems in their biophysical complexity and interaction [57,58].

The multifunctional ESs assessment is at the base of sustainable land use planning, and it is the result of a composite index score that supports land-use suitability analysis, achieving a better environmental quality of land use transformations [59].

The aim of this study is to create a replicable framework of spatial environmental indicators that can be used in a Geographic Information System (GIS) to design a local GI. We decided to select as the Area of Interest (AoI) the Çiğli district, Izmir (Turkey), one of the most dynamic and environmentally and socially heterogeneous districts of İzmir. According to the most recent bibliography on this issue [47,60,61], a set of open-access digital environmental layers were superimposed to analyze the system and obtain a fully replicable methodology for GI design. The layer superimposition served to identify all those green areas that can be included in an extended and organic green network at the district level which includes core, edges, and connection areas. The contribution of this article wants to open up new methods used by researchers, professionals, and other stakeholders to increase the digital biophysical knowledge of urban areas to design multifunctional GI. Finally, the outputs will help identify data needs and knowledge gaps, the research being of an extremely experimental character.

## 2. Materials and Methods

The methodology of this study is structured as follows: we first analyzed the problem of urbanization in the district while estimating the potential environmental effects. We focused on the observed changes between agricultural areas into the urban texture or commercial and industrial units as the most dangerous for biodiversity and the ecological structure.

We first analyzed the context by using the Copernicus Urban Atlas dataset from the most recent release (2018). We calculated the land use compositions and their distributions in the district. Then, we performed a Land Use Change (LUC) observation through GIS modelling to understand the most recent trends. LUC is empirically measurable by the superimposition of identical land-use datasets of (at least) two different temporal thresholds (2012–2018). Once intersected, the calculation of the features that changed their land destination has been used to track and quantitatively report the land use transformation.

Then, we analyzed and superimposed by GIS processing three different layers as representative of two fundamental categories of Ecosystem Services: the Copernicus High-Resolution Imperviousness layer, the Copernicus Tree Cover Density Layer (both selected as a proxy of the regulative capacity), and the Habitat Quality which has been produced by using InVEST software (as a proxy of the supporting capacity).

Impervious surfaces have long been observed as a measure of the urban environment's degradation, and with the increasing trend of urban expansion, they've become a major threat to habitat health [62]. Moreover, urban areas, especially in İzmir, have a higher intensity of impervious surfaces [63].

The tree cover density degree, coupled with the imperviousness, can describe where and how much open space there is in a specific catchment and the vegetation density of these green areas.

Finally, the Habitat quality of Çiğli and the impact of urban areas on habitat were analyzed with the open-access software Integrated Evaluation of Ecosystem Services and Trade-off (InVEST) [64] while employing Land Use and Land Cover (Urban Atlas 2018) data and the Imperviousness layer as inputs.

We then post-processed the layers by employing an unweighted overlay and hotspot analysis to define the critical thresholds of the new composite values and finally, we operate some spatial renderings to design the core, edge, and connection areas of the GI (Figures 1 and 2).

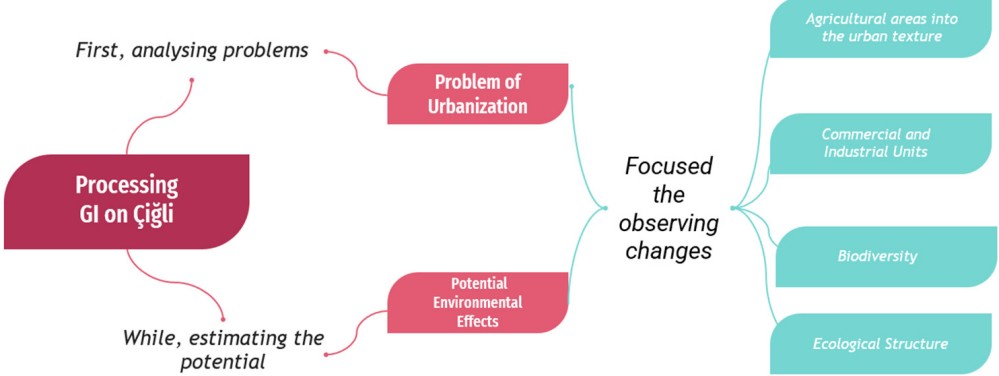

**Figure 1.** Logical workflow (source: author's elaboration).

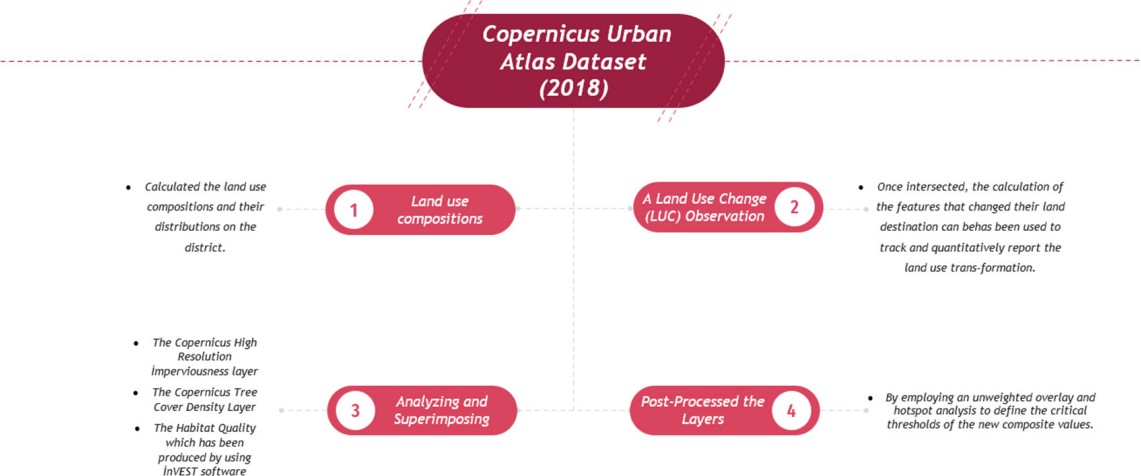

**Figure 2.** Analytical workflow of the GIS processing phases (source: author's elaboration).

### 2.1. Study Area and Context

Çiğli is a district in the metropolitan city of İzmir (Turkey), located in the northern part of the bay of the third-largest city in Turkey. The north part of the city has been historically developed as an industrial zone [65,66]. The district spans 13.900 hectares, including a heterogeneous land-use configuration and a varied landform. There is Sasalı Natural Life Park in the district, one of the largest seminatural zoos in Turkey, a military airport, a regional railway station, and many recently-developed urban areas with different densities. The district landform is characterized by the plain formed by the old Gediz River delta between the mountains on the north and the Gulf of İzmir on the south. The district has a wide coastal plain area that is barren and marshy and surrounds the Gulf. The İzmir Çanakkale highway creates a great barrier between the mountainous and plain land. According to Urban Atlas dataset of 2018 [67], the most abundant part of the area is

covered by natural areas (36%) and urban areas (34%). Green urban areas occupy less than 1%, and water bodies occupy 16% of the AoI (Figure 3).

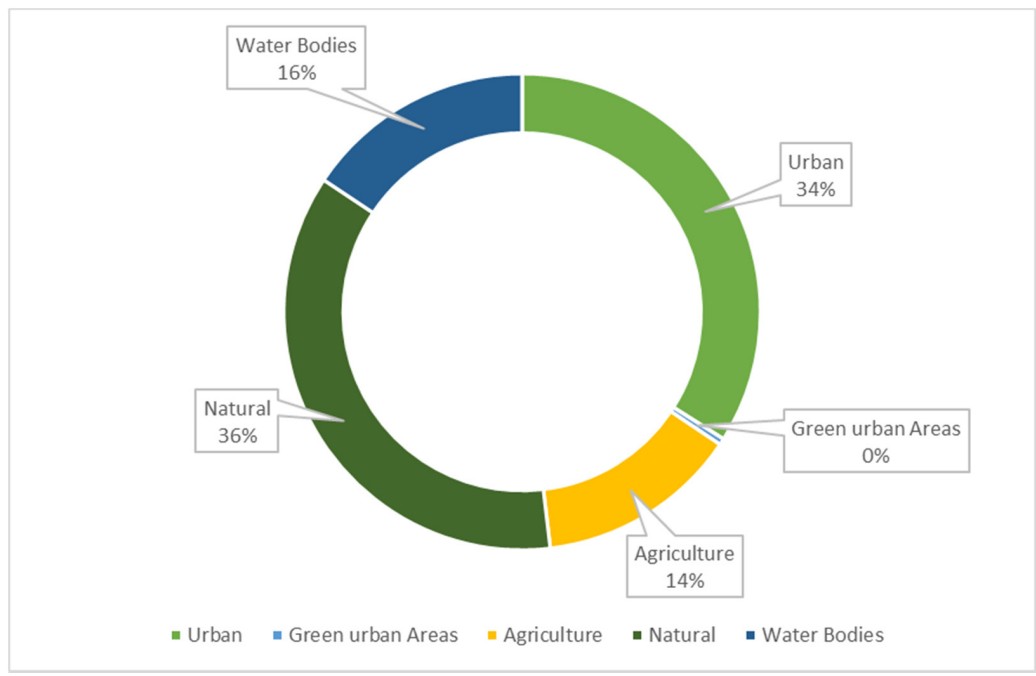

**Figure 3.** Land Use Land Cover composition in the Area of Interest (source: author's elaboration).

### 2.2. Land Use Change Analysis

As introduced, the land-use change analysis has been used as preliminary basic information to understand what has happened in recent years in the selected district and generate an initial assessment of the ongoing environmental trends. To empirically analyze these land substitutions, land use maps that belong to two different temporal thresholds were processed by downloading the original vector data format from the Copernicus Land Services (Urban Atlas 2012 and 2018).

Changes were analyzed spatially, visually and numerically, thus discussing, understanding, and visualizing problems to find solutions. The two Urban Atlas datasets of 2012 and 2018 were clipped with the district borders of the study area, and then processed by intersection using ESRI ArcGIS (ver. 10.8). The post-processing phase has been contradistinguished by an analysis of the attribute table using the .dbf file. The numerical computation of land-use change has been conducted through Microsoft Excel (pivot function). Then, a cross-tabulation matrix was produced and analyzed.

The cross-tabulation matrix has been used to calculate land use transition differences. Lastly, land use composition was then checked. Data were collected under five main land domains: urban, green urban areas, agriculture, nature, and water. It has been observed how these areas have decreased and increased from 2012 to 2018.

### 2.3. Analysis of Imperviousness

As underlined by many authors, the process of urbanization produces numerous changes in the natural environments it replaces [68,69]. Moreover, all urbanization processes are characterized by different degrees of imperviousness (not all urban areas are equally impermeable). Thus, we deepen the land-use change analysis with the imperviousness degree assessment. The major impacts of impermeabilization include habitat fragmentation and changes to both the quality and quantity of the stormwater runoff and resulting in changes to hydrological functioning [70,71].

The analysis of this index has been carried out while employing the raster statistic function to the original High-Resolution imperviousness raster layer provided by the Coper-

nicus Services (20 m ground resolution). The tool provides a computation for each future of a vector layer for all the statistics from an original raster layer (average imperviousness degree for each land use type).

### 2.4. Analysis of Tree Cover Density

Urban forests and vegetation, in general, are important hotspots of biodiversity, even in densely built urban areas [72,73]. In addition, peri-urban forests placed on urban fringes can generate important benefits for citizens in terms of biodiversity (greater tree species diversity, higher tree age, and greater share of deciduous species) but also for the many regulative functions that natural hotspots can generate (evapotranspiration, water and air filtering, cooling effect) [39,74]. The latest advancements in afforestation processes in dense urban areas include tiny forests and vegetation densification in small private plots as well. The primary goal of urban forestry planning and management is reaching at least a minimum level of tree cover density. Accurate measures of tree cover density are required to know where the distribution and the quality of densely vegetated natural areas are mainly concentrated. For this purpose, we employed the Tree Cover Density index to understand the distribution and the quantity of forested areas in the district.

Analogous to the imperviousness degree, the analysis of this index has been carried out while employing the raster statistic function, thus obtaining the average tree cover density for each land-use type in the AoİI.

### 2.5. Modelling the Habitat Quality

The Habitat Quality model requires three main inputs: a Land Use Land Cover map, a map of the habitat threats (e.g., anthropic areas), and a sensitivity table that associates the sensitivity to the threats for each land-use class and the degree of potential biodiversity. Once the Land Use Land Cover raster file has been generated, transforming the Urban Atlas 2018 digital map into a raster using the polygons to raster tool, the Imperviousness layer was selected as the main threat for the model. The Habitat Sensitivity table has been computed while considering various sources of information on the habitat quality associated with several types of land uses and employing the Tree Cover Density statistics in the district as a proxy for the biodiversity (Habitat) value in the sensitivity table. The more a habitat has a high value, the more this habitat is sensitive to anthropogenic sources of threats (Urban).

The sensitivity table for each habitat type was then created, whose numerical content is the score of the average distribution of the imperviousness layer for land-use classes (Table 1).

**Table 1.** Sensitivity Table of the Habitat Quality model.

| Lulc | Name | Habitat | Urban |
|------|------|---------|-------|
| 1 | Forests | 1 | 1 |
| 2 | Continuousu | 0.1 | 0 |
| 3 | Discontinuous | 0.5 | 0 |
| 4 | Industrial | 0 | 0 |
| 5 | Discontinuouslow | 0.5 | 0.5 |
| 6 | Land withoutcu | 0.5 | 0.5 |
| 7 | Arable land | 0.6 | 0.4 |
| 8 | Mineral extraction dump | 0.1 | 0 |
| 9 | Permanent crops | 0.7 | 0.7 |
| 10 | Pastures | 0.8 | 1 |
| 11 | Herbaceous vegetation | 0.8 | 1 |
| 12 | Isolated structures | 0.6,0.2 | |
| 13 | Other roads | 0.3 | 0 |
| 14 | Railways | 0.3 | 0 |
| 15 | Fast transit roads | 0.3 | 0 |
| 16 | Discontinuousmedi | 0.5 | 0.3 |

**Table 1.** *Cont.*

| Lulc | Name | Habitat | Urban |
|------|------|---------|-------|
| 17 | Airports | 0.3 | 0 |
| 18 | Green urban areas | 0.5 | 0.5 |
| 20 | Water | 1 | 1 |
| 21 | Open spaces | 1 | 1 |
| 22 | Wetlands | 1 | 1 |
| 23 | Discontinuousvl | 0.5 | 0.5 |
| 24 | Construction sites | 0.3 | 0 |
| 25 | Sports | 0.3 | 0 |

As for the threat map, we decided to use the imperviousness footprint as the only source of anthropogenic disturbance for the habitat while assigning a maximum distance of 800 m. and a linear decay function. The relative excel file (threat.csv) was created where necessary information such as the name of the threat, its range of effects, and its weight has been inputted numerically.

## 3. Results

### 3.1. The Environmental Characteristics of Çiğli

The Çiğli district comprises 36% natural areas, 34% urban areas, 16% water bodies, and 14% agricultural areas. The land-use change transition observed between 2012 and 2018 showed that the highest increase concerns the construction sites by 79%. The most common land-use type is isolated structures, with a decrease of 21%. Particularly, while analyzing the trends of change at the land-use sub-classes, increasing trends were observed for construction sites, discontinuous dense urban fabric, discontinuous urban fabric, continuous low-medium density, dense urban fabric, fast transit roads and associated land, green urban areas, industrial, commercial, public, military and private units, land without current use, mineral extraction and dump sites, and other roads and associated land. On the other hand, decreases were observed for arable lands (annual crops), discontinuous very-low-density urban fabric, forests, herbaceous vegetation associations (natural grasslands moors), isolated structures, open spaces with little or no vegetation (beaches, dunes, bare rocks, glaciers), pastures, and permanent crops (vineyards, fruit trees, olive groves). Water, airports, railways and associated lands, and sports and leisure facilities are the land-use typologies that have not changed in the Çiğli region between 2012 to 2018.

When we look at the changes in land-use macro classes, water bodies decreased by 3%, natural areas decreased by 25%, and agriculture decreased by 24% from 2012 to 2018. There was an increase of 49% in urban areas and 1% in green urban areas. Agricultural areas have been steadily decreasing, whereas urban areas have rapidly increased.

Çiğli displays a great variability in the imperviousness index, ranging from the maximum value (quantity of sealed surface calculated as 100%) recorded, which is 100%, and the minimum value, recorded as 0% (Table 2). The mean imperviousness of the district is 10.98%, which can be considered a relatively low index for İzmir's metropolitan area (the average imperviousness of the districts in İzmir's Province is 21.17%) and the eastern part of the AoI displays a higher concentration of sealed surfaces due to the existence of the core built-up areas (Figure 4). The western and southern parts of the land are mostly covered by agricultural, marshy, and sandy areas which are not sealed by anthropic surfaces.

**Table 2.** Statistics of the Imperviousness and Tree Cover Density degree datasets.

| Indıcator/Index | Max | Min | Mean | Units |
|-----------------|-----|-----|------|-------|
| Imperviousness | 100 | 0 | 10.98 | % |
| Tree Cover Density | 100 | 0 | 13.86 | % |

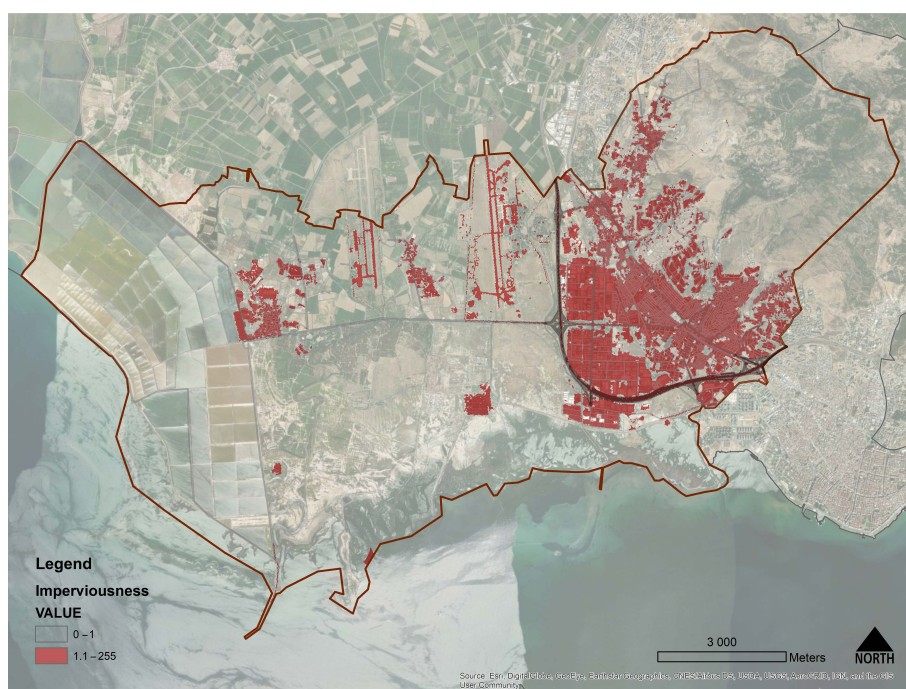

**Figure 4.** High-Resolution Imperviousness Degree on Çiğli district (source: author's elaboration).

Analogously to the Imperviousness degree, the district records the maximum values of the quantity of tree cover density (100% tree cover density, recorded in the northeastern mountainous, forested areas), whereas the average value is 13.86% (Table 2). The same average value for the entire districts of İzmir's Province records a value of 16.48%, indicating that even if the quantity of open, permeable spaces is above the average, the quality and the degree of vegetation density in Çiğli are relatively low. Indeed, the mean value also indicates that the average tree cover in this portion of the metropolitan area is quite low because of the presence of the Gediz Delta River, which created a plain fertile area used mostly for intensive agricultural purposes (Figure 5). This numerical consideration demonstrates the district's potential utilization of green densification processes.

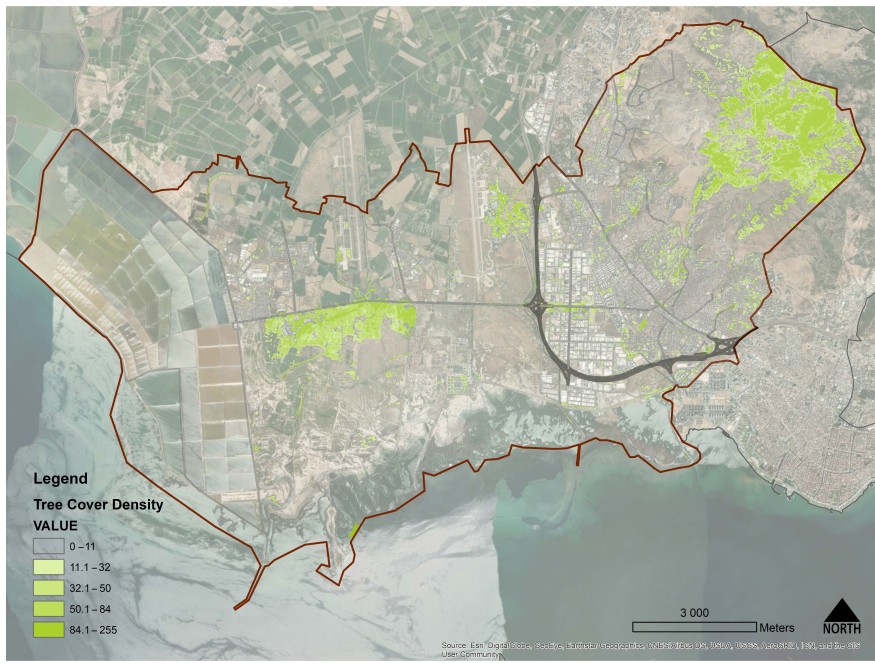

**Figure 5.** Tree Cover Density Degree on Çiğli district (source: author's elaboration).

Looking at the indicator's distribution, there are very few densely vegetated areas and the density of trees increases towards the northeast.

### 3.2. Modelling Output

The Habitat Quality output layers represent the biophysical spatial distribution of the potential biodiversity of Çiğli and the spatial distribution of threat values for the same catchment (Figures 6 and 7). Considering the sensitivity of habitats to the threats, we can observe that the most sensitive land uses are forests, open areas, wetlands and water, with a value of 1. The second sensitive land use is the herbaceous vegetation, then the pastures (with a value of 0.8), and the permanent crop (with a value of 0.7). Obviously, the hardly sealed land use classes such as industrial, continuous fabric, and mineral extraction dumpsites have no sensitivity to threats since they do not represent any habitat.

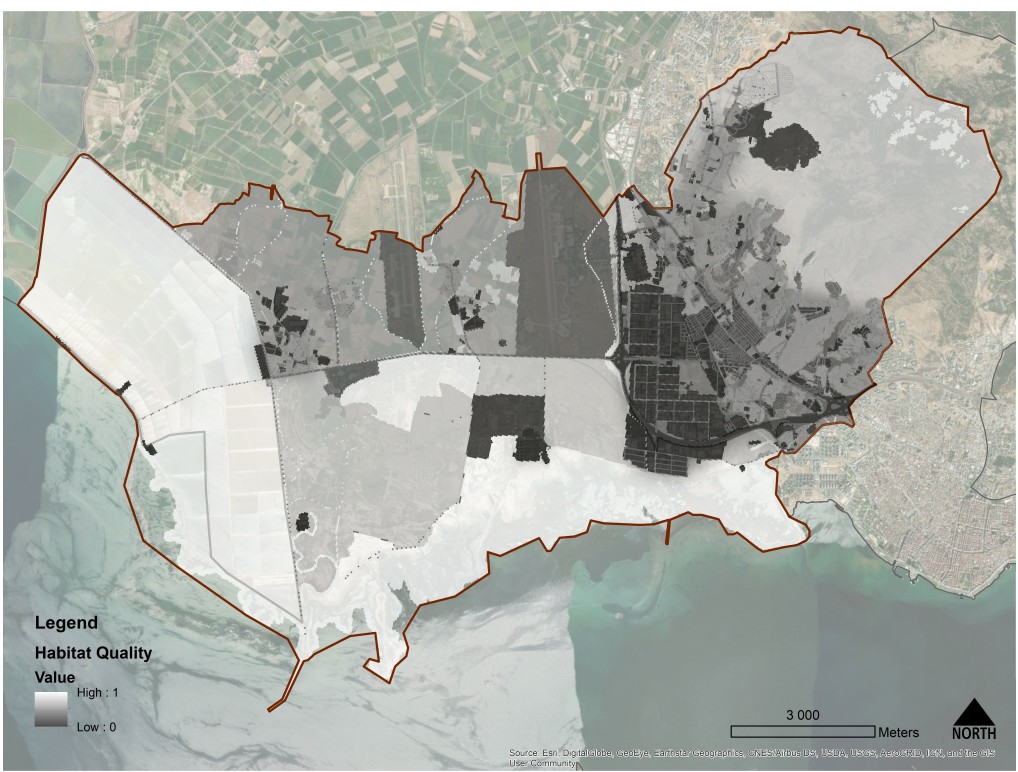

**Figure 6.** Habitat Quality values of Çiğli district (source: author's elaboration).

The interaction between the habitats and the impervious surfaces shows how the (i) distribution of habitat values is spatially clusterized in the district with a concentration of high values (0.75 average pixel value) on the south and western borders, average distribution of biodiversity on the central part of the AoİI (0.60 average pixel value), and then the low-quality cluster of the densely settled area (0.58 average pixel value) and (ii) how the urban areas affect the habitat quality at the urban fringes and how they fragment the landscape. Figure 6 clearly shows how the quality of natural and seminatural areas exponentially decreases while approximating to the urban fringes while emphasizing the ecological edge effect (the average degradation pixel index rises from 0.20 to 0.90 in a couple of hundred meters).

When the Habitat Quality of Çiğli district has been examined, it is seen that the average value is low (0.48) if compared to the average provincial district's distribution (0.61), but is higher if compared to the average values of the districts which are part of the dense urban agglomeration of İzmir (0.31). Moreover, habitat decay of Çiğli displays a lot of White-coloured areas which are not suitable for habitat. Fragmentation and impermeability are concentrated in these areas while representing a barrier to the ecological continuity of the district.

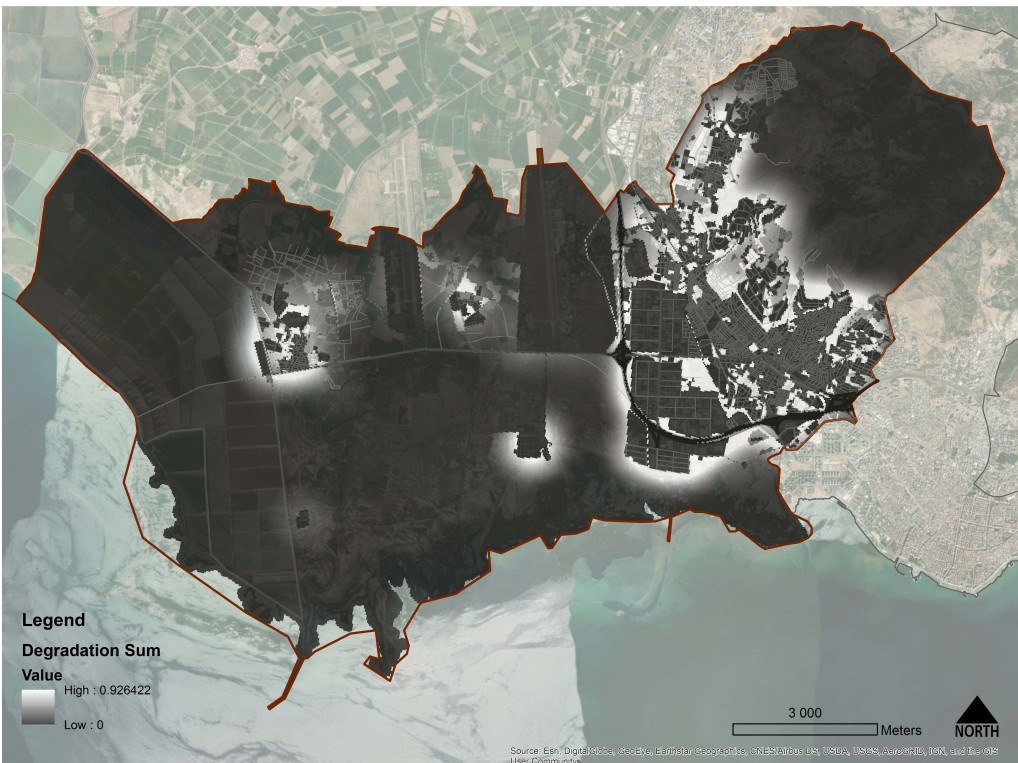

**Figure 7.** Habitat Degradation values of Çiğli district (source: author's elaboration).

### 3.3. Aggregating the Layers

As anticipated, this research aims to use the potential utilization of open-source datasets to design a GI in the district. To do so, Tree Cover Density, Imperviousness, and Habitat Quality maps were eventually spatially superimposed as input layers for a weighted overlay analysis. The above-mentioned layers have been used to set a new composite index which will be discussed in the light of the multifunctional distribution of supporting, regulative and aesthetic ESs in the catchment.

Composite indicators produced by overlaying techniques are quite common in applied environmental research and design. From McHargh to now, the distribution of composite spatial layers has been used to set ecological, environmental, and landscape measures to mitigate the anthropization processes and augment the potential biodiversity and connectivity of green urban areas [75,76]. Among the various techniques of aggregation, the Ecosystem Service Capacity has generally considered a technique which produces a pixel distribution of the multi-systemic land suitability, here intended as the land's capacity to deliver multiple ESs simultaneously. A spatial aggregation of a set of indicators (input layers) that represent the different "dimensions" of a phenomenon to be measured is called "composite index", and can be obtained through the spatial mathematical combination of the original inputs layers [77,78].

As a result of the weighted overlay (Figure 8), we discovered a predominant concentration of higher composite environmental values on the district's northeast and west sides (Figure 6). Still, numerous high composite values are present in the densely settled area of the district while demonstrating that urban multifunctional ecosystem services can be delivered by urban porosities and by public and private green areas that produce biodiversity and regulate the microclimatic conditions. Although there is a common understanding that even the urban areas can host some biodiversity while furnishing ecosystem services on the private, green, and porous surfaces of gardens, these densely inhabited spaces are the most eco-systemically-vulnerable while showing a low capacity degree of multifunctional environmental values.

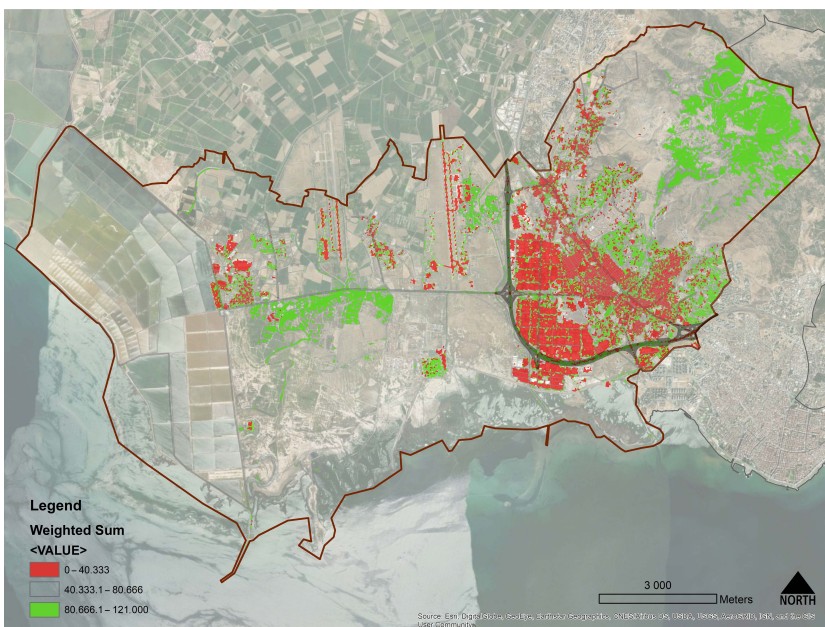

**Figure 8.** Weighted Sum Result (source: author's elaboration).

To facilitate and standardize the recognition and discussion of these multifunctional areas, we made a final post-processing phase that included the statistical spatial treatment of the weighted overlay. Specifically, we applied a Hotspot analysis with a threshold distance of 500 m and extracted all positive values (Gi-scores 1, 2 and 3). This technique allowed us to extract and distinguish at least three elements of the GI: the multifunctional core areas (Gi-values = 3) and the edge areas (Gi-values = 1 and 2), plus the urban links which are represented by all the narrow, fragmented urban areas that potentially can be re-designed as biodiversity corridors in the densely built-up system (Figure 9). Patch identification is based on the input layer aggregation method (Unweighted Overlay) and its clusterization method (HotSpot Analysis), which relies on the technical characteristics of layers; in general, reducing grain size (i.e., 10-m vs. 100-m cells) will increase the number of patches that are identified during cluster analysis because additional detail is resolved on the finer scale [79–81].

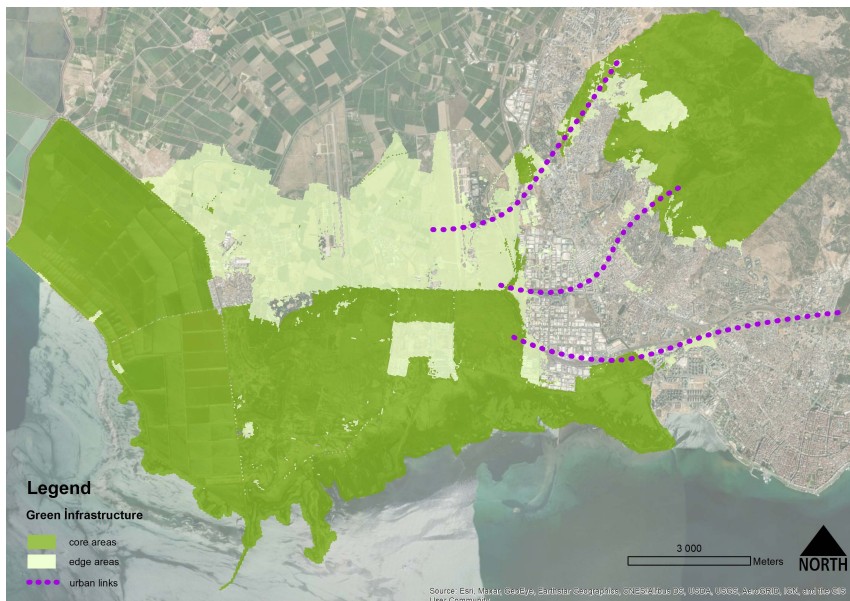

**Figure 9.** The Green Infrastructure design (source: author's elaboration).

## 4. Discussion

GI design is a process whose goal is to evaluate the multifunctional character (supporting, regulative, provisioning, and recreative) of the land and define the preservation, transformation, and spatial regeneration strategies to enhance the biodiversity and guarantee a more healthy, safe, and resilient environment in cities. As mentioned in the result section, the weighted sum result shows how the multifunctional character of the land is scarcely present in two different clusters. These clusters are mostly composed of high-density/mostly impermeable residential areas. At the same time, the analytical outputs demonstrate that there is a great amount of agricultural land that can be used for more sustainable and resilient agro-ecological practices, especially in those rural areas placed at the interface between the eastern densely developed settlement system and the more seminatural environment that is composed by the sedimentary basin of the Gediz Delta. Unfortunately, even if the western part of the district is composed of agricultural and seminatural uses, the presence of two airports, a water treatment plant, and other anthropic functions (Zoo) are reducing ecological connectivity. In these spaces, the re-introduction of more extensive and less water-demanding agriculture can be used to create a recreative, accessible environment for the local communities.

Nonetheless, the dense urban areas can be re-designed to create minor urban ecological connections, enhancing the internal micro-biodiversity or the space for residents' recreation and wellbeing. As basically argued by some bibliographies that focuses on urban biodiversity, the de-fragmentation of the densely inhabited built-up system is a priority for urban green infrastructure development [82,83].

### 4.1. Designing a Green İnfrastructure

According to the last enlightened approaches, urban planners should follow a basic principle while designing and regulating the space: "green where possible, grey only when needed" [30]. The built, densely inhabited environment can often offer micro-scale NBS opportunities, providing multiple benefits even in space constraints. Therefore, the integration of NBS in the urban landscape is a basic strategy to achieve the long term benefit of communities and citizens.

GI requires the formal integration of NBS into zoning schemes and their executive masterplans for their implementation and, therefore, requiring some basic spatial categorizations. Composite landscape metrics normally compute GI on a spatial dataset in which the input layers or the spectral data have been classified into some representative number of categories. An important assumption that must be recalled when interpreting these data is that spatial metrics classified into categories are presumed to be homogeneous within a given category.

Since the review by Gustafson (1998), the tools for analyzing landscape patterns have become well established, the behaviour of most metrics is well understood, and landscape metrics are widely used [84]. This work paved the way for understanding the heterogeneity of ecological systems (complexity and variability) both in time and space. This variability can be better understood by aggregated maps/layers that quantize variability by identifying relatively homogeneous patches.

Within this experimental approach, we wanted to promote the definition of the three elements recognized by a standardized GIS procedure for GI design (core, edge, and links). We aim to define the hierarchy of the environmental system during the design process and adequately support the decision making while taking strategic decisions in planning.

The following identification of three main GI typologies corresponds to the basic morphological spatial pattern analysis of landscape ecology elements in urban catchments, including core areas, islets, perforations, edges, loops, bridges, and branches. In general, larger heterogeneous natural and seminatural patches contain more species and often a greater number of individuals than smaller patches of the same habitat. At the same time, the relative abundance of edge and interior habitat affects species diversity within a patch. Finally, corridors can both add habitat and promote movement.

### 4.1.1. Strategies and NBS for Core Areas

Core areas are natural wetlands, grasslands, floodplains, urban forests, and coastal lagoons, all examples of ecosystems in and around urban areas that should be protected to secure existing benefits [72,79]. Urban forests are complex seminatural ecosystems located within urban vacant, open land or at the rural–urban interface, which provide strong biodiversity while lowering the vulnerability to climate change. Urban forests should be considered node landscape source areas that prevent ecological fragmentation and reduce biodiversity by connecting the settlements with the larger landscape eco-mosaic. The protection of existing natural resources in a city is a basic precondition to maintaining minimal functional and biodiversity values. Urban forest protection, reinforcement, and valorization are considered a priority in core ecosystem areas [85,86]. Additionally, re-naturalization, green densification, and valorization is a basic strategy to adapt to adversity and obtain the quick ability to adapt to hostile conditions and thrive with different odds. Design, landscaping, and replanting can enhance the tiny forest approach [87].

### 4.1.2. Strategies and NBS for Edge Areas

Edge areas are mostly composed of transitional rural ecosystems located between the core multifunctional ecosystem sites and the borders of the hardly-developed anthropic system. They play a vital role in decreasing the habitat edge-effect and can be used for recreative aesthetic functions asg their location is in direct proximity to the settlements [88].

Edge areas should be rehabilitated to enhance their performance, functioning, and benefits. These areas should be selected as a priority for future ecological compensation measures since the increase of their biodiversity can drastically reduce the edge effects in core habitats while safeguarding the biodiversity nodes of the system. Since edges are located in plain, fertile areas, urban farming and other compatible NBS with the rural productive character of this land can be prioritized. The importance of urban farming relies not only on the preservation of green spaces but also on their related socio-economic benefits. They increase access to the healthy, affordable, fresh product delivered at the community level. They encourage socialization and awareness (to know where food comes from).

Finally, the co-benefits of productive ecosystems can also be enhanced through anti-erosion and less hydro-demanding measures and the implementation of sustainable agro-forestry methods.

### 4.1.3. Strategies and NBS for Urban Links

As mentioned, even the smallest open space in a densely inhabited system can directly benefit from nature's multiple ecosystem capacities in urban areas. Opportunities also exist to enhance and strengthen the green corridors that connect the core and edge areas. Creating new NBS in link areas can mitigate impact and strengthen urban resilience. This includes micro capillary interventions such as green roofs, vegetated facades, constructed wetlands, and bioretention areas. These new NBS can also provide other co-benefits to local communities and create the movement of people between the catchment's urban, peri-urban, and rural landscapes.

### 4.1.4. Specific Interventions for the Existing Settlements

Considering the various general strategies of conservation and valorizations inside the GI, the district has some relevant problems regarding the existing stock of settlements; at the same time, potential sites can be prioritized to trigger a diffuse environmental regeneration program that includes GI development.

First, the İzmir Atatürk Organized Industrial Zone, located in the city's east end, is a problematic area because of its pollution and contamination of the surrounding rural system. Here, green industry projects and jeopardized afforestation pilot areas around the area can mitigate the spread of air pollution.

Moreover, to increase the habitat quality in the AOI, creating green areas around the Atatürk Organized Industry Stream and the canals in the district can be pivotal, as they can ensure green continuity and mitigate the landscape edge effect.

In general, diffuse interventions in grey areas are also extremely important. Integrating green walls, green roofs, green fences, and parklets in densely urbanized and sealed areas is crucial. Creating green rain gardens and public rain plazas, or any solutions to accumulate water with bioswales on agricultural areas rather than water filtering strips in the city with biofilters can be considered extremely beneficial for citizens' wellbeing. Finally, permeable roads, green noise barriers, and green vegetated strips can work diffusely as carbon sinks.

*4.2. Limits of the Study*

We know that implementation of GI exclusively based on remote-sensed data constitutes a purely technocratic approach to environmental planning, which contradicts the organicity, local understanding, and explorative capacity of design or co-designing the environment with key stakeholders.

It should be kept in mind that the employment of ecosystem models relies on sources that should always be updated. Normally, mapping processes that are based on numerical datasets should be checked in consultation with local experts to confirm the maps' reliability. In addition, the above mentioned GI design process has been performed using only three datasets. Still, composite environmental indicators and the multifunctional capacity of land may be calculated using much more input layers. Finally, this study has not been tested in the field or in any settled area. Indeed, we know that planning and implementing NBS in Izmir requires several special considerations. The city has distinct wet and dry seasons while representing a challenging site for plants, trees, and grasslands where NBS are located. Therefore, we suggest that the exact definition of NBS should be based on a detailed analysis of the microclimatic factors that influence the vegetation's health, such as solar radiation, rainfall patterns, surface water flows, and groundwater levels. The executive NBS design phase should also include the catalogue of indigenous plants and trees that present ideal growing conditions in the site and maintenance plans. Even if the costs of NBS are not included in this research, a real application of GI design should consider the resources for watering and maintenance for the long-term implementation of NBS.

## 5. Conclusions

This study proposed an easy and replicable methodology to design a Green Infrastructure at the neighbourhood level in the Çiğli district. The method we used combined historical land-use change analysis with environmental and ecosystem mapping. The layers (Imperviousness, Tree Cover Density, and Habitat Quality) were superimposed by overlay analysis (composite multifunctional indicator) and classified by HotSpot technique to obtain three GI categories: core, edge, and link areas.

This work constitutes only a general guide to support the process of GI co-design, which includes a much more detailed, expanded, and time-consuming process to analyze and develop solutions for NBS implementation. Nevertheless, we demonstrated that nowadays it is possible to set a basic framework for GI design using open-source data and ecosystem modelling in every part of the world. While doing this, we know that the site-specific detection of ecological values, habitats, and their degradation should be supported and compared by making more specific analyses. Still, the findings of the empirical modelling are consistent with the real ecological condition of the district. We empirically measured how the habitat quality of Çiğlin is concentrated in areas to the south and west where there is no urbanization, and it was observed that the habitat had been degraded in the east and north-west regions where urbanization is concentrated and in areas where water treatment plants are located. The GI was designed to support the decision-making process during urban planning processes, and it was envisaged that the nature-city balance in the district would be maintained. In comparison to grey infrastructure, NBS provides a variety of benefits to society, such as flood risk reduction, heat stress reduction,

human health, recreation, tourism, and biodiversity. The full suite of benefits that NBS offers is often an important factor in decision making. Not only achieving the nature-city balance but also minimizing the damage to the habitat and ecosystem in the district from urbanization has been foreseen with these recommendations.

**Author Contributions:** Conceptualization, S.S., B.E. and B.A.; methodology, S.S., B.E. and B.A.; software, B.E. and B.A.; validation, S.S.; formal analysis, B.E. and B.A.; data curation, B.E. and B.A.; writing—original draft preparation, S.S., B.E. and B.A.; writing—review and editing, S.S., B.E. and B.A.; supervision, S.S. All authors have read and agreed to the published version of the manuscript.

**Funding:** This research received no external funding.

**Institutional Review Board Statement:** Not applicable.

**Informed Consent Statement:** Not applicable.

**Data Availability Statement:** Not applicable.

**Conflicts of Interest:** The authors declare no conflict of interest.

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
