# Peer review of "Designing Urban Green Infrastructures Using Open-Source Data—An Example in Çiğli, Izmir (Turkey)"

_urbansci, doi:10.3390/urbansci6030042_

Round 1

Reviewer 1 Report

Manuscript ID: urbansci-1749706

Title: Designing Urban Green Infrastructures using open-source data – an example in ÇiÄŸli, Izmir (Turkey)

The topic and the idea of the study is very interesting and in line with the approach promoted nowadays in urban areas development However, the presentation of the study, including methods, results which are described in a very limited form, as well as discussion have many weaknesses.

Main comments and suggestions:

1. The title is well prepared.

2. Abstract is quite long and not so well constructed - after too much developed “general introduction” to the topic, there is no information about e.g. aim of the study, main research results, and main conclusion. It should be improved to inform the readers about the study but still in synthetic way.

Key words are well selected, related to the topic.

3. Introduction - is developed but mostly in the presentation of general aspects of GI, etc., at the same time some words are repeating so many times in next sentences, e.g. climate, climate change (lines 35-43, 48-53, etc.). Literature cited in the Introduction is quite old - there are accessible many new publications about the GI, NBS, also related to resilient socio-ecological and technological systems – it should be improved.

A weakness of the Introduction is not much developed background of the role and the use of methods proposed by Authors, especially for a kind of studies presented in the manuscript; there is also no clear explanation, why especially the aspects selected for the study are important and sufficient for proposed development model for ÇiÄŸli, Izmir, etc.

The aim of the study is not presented, it is just a part of continuous description with some relations to background, methods and explanation repeated from the Introduction (?) (lines: 98-111). It must be improved.

5. Materials and Methods – the division into subsection is correct, however - as above-mentioned - the description of research stages and their scope must be more factual and clear; generally first sentences in most subsections in this part should be rather a part of Introduction.

Figure 2 - it is not clear why it is presented? Also the description is not clear. Generally description of all Figures must be improved.

6. Results – the description of all analyses and studied aspects is rather chaotic and not deep, there are short presentations and  there is not much data included.

(also Table 1 and 2 should be presented as one, it is not clear why Authors create tables just with only one line?).

Especially description of subsection 3.3. Aggregating the layers is weak. Authors inform that “The post-processed layer has been used to set a new composite index which will be discussed in the light of the multifunctional distribution of supporting, regulative and aesthetic ESs in the catchment.” (lines 270-272) – but there is no description of the proposed composite index (?). The presentation of results of ‘weighted overlay’ is very limited, and thus much information is missed. The study is very general and ‘sketchy’ in my opinion.

7. Discussion – regarding the lack of data presented in Results, this part focuses mostly on some general repeated information of GI (lines 279-303), but is not much developed in relation to obtained results and their value/role.

A part of this section is rather interpretation of some results (lines 294-311), and should be moved to the section of Results.

'The Green Ä°nfrastructure design' – which is also a kind of result in my opinion – is very (too) general, and thus shows the weakness of the study, mainly the lack of details, which is consistent with the limited presentation of results. Regarding those missed information, also the proposed Strategies cannot be well argued.

8. Conclusions – most of information is general and repeated.

9. Others:

- the quality of Figures is low

- language needs improvement

Summing up, the weak form of presentation and general low level of details/data is insufficient and limits the scientific value of the research. I cannot recommend the manuscript for publication in its present form.

Author Response

The topic and the idea of the study is very interesting and in line with the approach promoted nowadays in urban areas development However, the presentation of the study, including methods, results which are described in a very limited form, as well as discussion have many weaknesses.

Thank you so much for these valuable comments.

Main comments and suggestions:

  1. The title is well prepared.

  1. Abstract is quite long and not so well constructed - after too much developed “general introduction” to the topic, there is no information about e.g. aim of the study, main research results, and main conclusion. It should be improved to inform the readers about the study but still in synthetic way.

Thank you for your observation. Now we reduced the abstract's length while emphasizing the study's aims and clarifying the results achieved by this study.

Key words are well selected, related to the topic.

Thank you for this observation.

  1. Introduction - is developed but mostly in the presentation of general aspects of GI, etc., at the same time some words are repeating so many times in next sentences, e.g. climate, climate change (lines 35-43, 48-53, etc.). Literature cited in the Introduction is quite old - there are accessible many new publications about the GI, NBS, also related to resilient socio-ecological and technological systems – it should be improved.

Thank you so much for your observation, we tried to re-arrange the Introduction considering your valuable comments. We removed the repetitions and updated the references.

A weakness of the Introduction is not much developed background of the role and the use of methods proposed by Authors, especially for a kind of studies presented in the manuscript; there is also no clear explanation, why especially the aspects selected for the study are important and sufficient for proposed development model for ÇiÄŸli, Izmir, etc.

Thank you so much for your observation, we integrated the Introduction emphasizing the background of Gİ design and how the concept evolved while becamıng crucial for climate change (especially for coastal cities like İzmir)

The aim of the study is not presented, it is just a part of continuous description with some relations to background, methods and explanation repeated from the Introduction (?) (lines: 98-111). It must be improved.

Thank you for your observation. We clearly declared the aim of the study.

  1. Materials and Methods – the division into subsection is correct, however - as above-mentioned - the description of research stages and their scope must be more factual and clear; generally first sentences in most subsections in this part should be rather a part of Introduction.

Thank you so much for this observation. We tried to introduce the workflow properly in the opening of the methodology section and remove the more general, introductive parts from the methodology. We hope this new version gain in clarity.

Figure 2 - it is not clear why it is presented? Also the description is not clear. Generally description of all Figures must be improved.

We changed figure 2 and we revised the captations of all the figures

  1. Results – the description of all analyses and studied aspects is rather chaotic and not deep, there are short presentations and there is not much data included.

Thank you for your observation, we tried to present more scientifically the results of the study, especially using some comparative results with the same indicators at the provincial level.

(also Table 1 and 2 should be presented as one, it is not clear why Authors create tables just with only one line?).

Thank you for your observation, now we merged the tables.

Especially description of subsection 3.3. Aggregating the layers is weak. Authors inform that “The post-processed layer has been used to set a new composite index which will be discussed in the light of the multifunctional distribution of supporting, regulative and aesthetic ESs in the catchment.” (lines 270-272) – but there is no description of the proposed composite index (?). The presentation of results of ‘weighted overlay’ is very limited, and thus much information is missed. The study is very general and ‘sketchy’ in my opinion.

Thank you so much, we tried to give more scientific concreteness to our chapter while introducing the multifunctional ecosystem capacity expressed by composite indicators.

  1. Discussion – regarding the lack of data presented in Results, this part focuses mostly on some general repeated information of GI (lines 279-303), but is not much developed in relation to obtained results and their value/role.

A part of this section is rather interpretation of some results (lines 294-311), and should be moved to the section of Results.

Thank you for this observation. We revised the discussion by taking into account these suggestions

'The Green Ä°nfrastructure design' – which is also a kind of result in my opinion – is very (too) general, and thus shows the weakness of the study, mainly the lack of details, which is consistent with the limited presentation of results. Regarding those missed information, also the proposed Strategies cannot be well argued.

Thank you so much for this observation. We worked diffusely in this section to give more concreteness to the methodology and explain the landscape metrics approach we used to define core, edge and link areas. We also modified the description of each cluster while being more coherent. We concentrated the parts on the limits in the appropriate section 4.2.

  1. Conclusions – most of information is general and repeated.

Thank you for your observation, we re-designed the conclusion and moved the study's novelty in this final part.

  1. Others:

- the quality of Figures is low

Thank you for this observation, we changed bad quality figures and exported alla the jpeg in A3 format with 300 dpi.

- language needs improvement

Thank you for this comment. We revised the sentencing entirely while taking care of the readability, and we also asked our Academic Writing Centre to make an extended final revision of the manuscript.

Summing up, the weak form of presentation and general low level of details/data is insufficient and limits the scientific value of the research. I cannot recommend the manuscript for publication in its present form.

We sincerely hope this new, fully revised version meets the criteria you mentioned

Reviewer 2 Report

1.         In general, the objective of this paper is not quite clear.  It will be appreciated if the authors could clearly describe the major contribution of this research in the Abstract and Conclusion sections. Furthermore, based on the results obtained, it will be much better for the author to provide some valuable recommendations to assist the urban city development.

2.         The authors studied the historical land-use change of an urban city, including the land cover, tree cover density, and imperviousness. However, I recommend the authors add a figure to present the overall research framework of this study for easier understanding of the readers.

3.         Please revise Figure 2 to a Table. In addition, please describe the meaning of those sensitivty values.

4.         The intention of Figure 3 is unclear. Please describe the meaning of the Imperviousness value.

5.         Please move the Table caption to above the Table.

6.         Table 1 and Table 2, if the Imperviousness and tree cover density is a percentage value, please explain those values shown in Figures 3 and 4)

Author Response

  1. In general, the objective of this paper is not quite clear. It will be appreciated if the authors could clearly describe the major contribution of this research in the Abstract and Conclusion sections. Furthermore, based on the results obtained, it will be much better for the author to provide some valuable recommendations to assist the urban city development.

Thank you so much for the valuable comment. We entirely reviewed the abstract and conclusion section, emphasizing the study's aim and how these achievements can practically support the land use planning process.

  1. The authors studied the historical land-use change of an urban city, including the land cover, tree cover density, and imperviousness. However, I recommend the authors add a figure to present the overall research framework of this study for easier understanding of the readers.

Thank you so much for your valuable comment, we now added a flowchart which explains the methodological process.

  1. Please revise Figure 2 to a Table. In addition, please describe the meaning of those sensitivty values.

Thank you for your observation. We changed accordingly.

  1. The intention of Figure 3 is unclear. Please describe the meaning of the Imperviousness value.

Thank you so much for your comment. We deeply modified the methodological description while trying to explain clearly the utilization of the different indexes (including imperviousness).

  1. Please move the Table caption to above the Table.

Done. Thank you.

  1. Table 1 and Table 2, if the Imperviousness and tree cover density is a percentage value, please explain those values shown in Figures 3 and 4)

Thank you for your comment. We deeply revise the results section while clarifying the distribution of the indexes and providing additional comparative data for their interpretation.

Reviewer 3 Report

I appreciate the interesting topic, quality methodology, and plenty of results. I have a few comments about the article:

Discussion

·         What is the importance of Urban farming? 

Conclusions

·         Draw the conclusion according to the created hypotheses and goals. 

English Errors:

Line 28: the paucity of digital data strongly limits this kind of empirical modelling-prcesses

Lines 85 and 88: e.g – the abbreviation e.g seems to be incorrectly punctuated. The right form – e.g.

Line 144/155: Dataset – Datasets

Line 196: Fifure 2 Figure 2

Line 209: …continuous low medium dense dense urban fabric… It appears you typed dense twice in a row

Line 288: spaces are the most ecosystemically-vulnerable while showing…I would write: these densely inhabited spaces have the most ecosystem vulnerability while showing....

Line 326: masterplan – master plan

Line 404: Dataset – Datasets

Line 413: in the site – on the site

Author Response

Discussion

  • What is the importance of Urban farming?

Thank you so much for this observation, we enphasized the importance of Urban Farming.

Conclusions

  • Draw the conclusion according to the created hypotheses and goals.

Thank you so much, we re-designed the conclusion accordingly

English Errors:

Line 28: the paucity of digital data strongly limits this kind of empirical modelling-prcesses

Lines 85 and 88: e.g – the abbreviation e.g seems to be incorrectly punctuated. The right form – e.g.

Line 144/155: Dataset – Datasets

Line 196: Fifure 2 – Figure 2

Line 209: …continuous low medium dense dense urban fabric… It appears you typed dense twice in a row

Line 288: spaces are the most ecosystemically-vulnerable while showing…I would write: these densely inhabited spaces have the most ecosystem vulnerability while showing....

Line 326: masterplan – master plan

Line 404: Dataset – Datasets

Line 413: in the site – on the site

Thank you for all your detailed indications, we made a comprehensive research of similar mistakes in the body of the text.

Round 2

Reviewer 1 Report

Manuscript ID: urbansci-1749706

Title: Designing Urban Green Infrastructures using open-source data – an example in ÇiÄŸli, Izmir (Turkey)

Authors: Stefano Salata, Bensu ErdoÄŸan, Bersu AyruÅŸ

I appreciate all works of the Authors, the manuscript, especially the presentation of the study, is well prepared and much better organized, thus its scientific soundness is higher.

Last comments and suggestions:

1. Abstract is quite well constructed however very general regarding much data presented in the study.

Key words are well selected, related to the topic.

2. Introduction - is much better organized in the context of its order. The aim of the study is more highlighted, including its argumentation.

However, in this paragraph Authors used some ‘general’ wording, e.g. in lines 142-143: ‘According to the most recent bibliography on this issue, a set of open-access ....’ – this ‘recent bibliography’ is needed to be cited here (and/or repeated from Introduction) – otherwise the readers will not sure which literature items are exactly about; this ambiguity must be resolved.

3. Materials and Methods – the division into subsection is correct, also the presentation/description of research stages and their scope is more clear in present form, easier to understand.

4. Results – the description of presented analyses, and also results is much better presented and more deep, thus more easy to understand by readers. The material presented in subsection 3.3. Aggregating the layers is clear in its present form. Figures are in their right positions and support the text.

5. Discussion – regarding the improvement of the section of Results, the section of Discussion is better organized, also main relations to obtained results are highlighted.

Generally the main direction of interpretation presented in Discussion is valuable, however the scope of literature cited in this section could be still more developed.

8. Conclusions – are not so deep, but related to the study presented in the manuscript, and explain what the study brings.

9. Others:

- Figures – it would be very useful to know, if the figures are elaborated individually by Authors and/or based on selected sources? this short information should be added to the figures’ description.

Author Response

I appreciate all works of the Authors, the manuscript, especially the presentation of the study, is well prepared and much better organized, thus its scientific soundness is higher.

Thank you so much for this general comment

Last comments and suggestions:

  1. Abstract is quite well constructed however very general regarding much data presented in the study.

Thank so much for this comment. We made some minor changes in the abstract in the suggested direction.

Key words are well selected, related to the topic.

Thank you for your comment

  1. Introduction - is much better organized in the context of its order. The aim of the study is more highlighted, including its argumentation.

However, in this paragraph Authors used some ‘general’ wording, e.g. in lines 142-143: ‘According to the most recent bibliography on this issue, a set of open-access ....’ – this ‘recent bibliography’ is needed to be cited here (and/or repeated from Introduction) – otherwise the readers will not sure which literature items are exactly about; this ambiguity must be resolved.

Thank you so much for this observation, we integrated the sentence with the relevant bibliography in the field.

  1. Materials and Methods – the division into subsection is correct, also the presentation/description of research stages and their scope is more clear in present form, easier to understand.

Thank you so much for your positive comment.

  1. Results – the description of presented analyses, and also results is much better presented and more deep, thus more easy to understand by readers. The material presented in subsection 3.3. Aggregating the layers is clear in its present form. Figures are in their right positions and support the text.

Thank you so much for your positive comment.

  1. Discussion – regarding the improvement of the section of Results, the section of Discussion is better organized, also main relations to obtained results are highlighted.

Thank you so much for your positive comment.

Generally the main direction of interpretation presented in Discussion is valuable, however the scope of literature cited in this section could be still more developed.

Thank you so much for this comment. We tried to emphasize this point in the discussion.

  1. Conclusions – are not so deep, but related to the study presented in the manuscript, and explain what the study brings.

Thank you for your comment.

  1. Others:

- Figures – it would be very useful to know, if the figures are elaborated individually by Authors and/or based on selected sources? this short information should be added to the figures’ description.

All figures were elaborated by the authors (the source has been added in the captation).